# Domain Switching-Based Nonlinear Coupling Response for Giant Magnetostrictive Materials

**DOI:** 10.3390/ma16144914

**Published:** 2023-07-09

**Authors:** Yunshuai Chen, Pengyang Li, Jian Sun, Guoqing Chen

**Affiliations:** School of Mechanical and Precision Instrument Engineering, Xi’an University of Technology, Xi’an 710048, China; chenyunshuai@stu.xaut.edu.cn (Y.C.); 1210211011@stu.xaut.edu.cn (J.S.); 1220211001@stu.xaut.edu.cn (G.C.)

**Keywords:** magnetostrictive material, nonlinear response, domain switching

## Abstract

This paper proposes a multilevel three-dimensional constitutive model based on a microscopically phenomenological approach from the domain rotation mechanism, which is a fully coupled self-consistent homogenization scheme considering the interactions between elastic–inelastic strain and hysteresis. Considering the interactions among magnetic domains, grains, polycrystalline complexes, and macroscopic phenomenology, we predict the nonlinear magnetostrictive response of Terfenol-D under different types of external force loads and magnetic excitations in various thermal environments involving multi-fields of coupled magnetic, elastic, thermal, and mechanical phenomena. The average values of the mechanical bulk strains for different magnetization states are obtained at the grain scale utilizing Boltzmann functions and a self-consistent homogenization scheme. A Taylor series expansion of the Gibbs function concerning the field variables and an adapted Jiles–Atherton model are used to construct the hysteresis coupled constitutive relations at the macroscopic scale. The results associated with the experiments show that the established model can reasonably predict the magnetostrictive response under different external mixed stimuli. It can provide theoretical guidance for the precise control of nonlinear vibrations and the optimal design of the rotating giant magnetostrictive transducers at both microscopic and macroscopic multiple scales.

## 1. Introduction

In modern industry, on the one hand, magnetostrictive materials are used in functional devices with high-precision displacement control based on quasi-static magnetic field driving with prominent strain characteristics [1,2,3,4], such as rotating giant magnetostrictive vibrating tool handles and transducers. On the other hand, they are used in functional devices with dynamic characteristics based on their fast response speed, wide frequency band, and low voltage drive characteristics [5], such as giant magnetostrictive thin-film actuators. Terfenol-D provides a more effective trade-off between higher strain and higher Curie temperature than other giant magnetostrictive materials. However, it has been found that, especially in ultrasonic vibration processing, functional devices frequently experience vibration instability during operation [6,7], i.e., the effect of external conditions such as force and temperature on the vibration characteristics of the constitutive material. Many efforts have been made to understand the causes of this unwelcome phenomenon. It is recognized that one of the critical causes is the strongly coupled nonlinear properties of the material of the drive itself due to its performance in the magnetic-elastic-thermal-force external environment [8,9]. As the magnetostrictive material is designed, a precise constitutive relationship plays a vital role in the study of the stability of dynamic characteristics and the promotion of high-performance applications.

From a macroscopic point of view, the development of the constitutive models has evolved from the initial linear ones to the nonlinear ones. However, these models are still incomplete in fully coupled magneto-elastic-thermal-mechanical behavior and hysteresis. At an early stage, based on magneto-mechanical interaction in a relatively small magnetic field region, Clark [10] proposed a linear piezo-magnetic model describing magnetostrictive rare-earth ferroalloys, and the simplicity of this linear treatment made the model widely used in engineering applications. However, in the operating environment of higher magnetic fields, the linear piezo-magnetic model can no longer accurately describe the nature of the magneto-elastic response of magnetostrictive materials. Subsequently, the intrinsic nonlinear phenomena of material excitation have been extensively investigated based on thermodynamic principles using the Taylor expansion of the Gibbs free energy with respect to some field variables, such as the SS model [11] (i.e., the standard square model), HT model [12] (i.e., the hyperbolic tangent model), and DDS model (i.e., the density of domain switching). However, it has been noticed that these models, while showing good nonlinear coupling agreement at low and medium applied magnetic fields, also show significant discrepancies between the predicted and experimental results at high magnetic fields and high prestress. Large magnetostriction driven by a high magnetic field is vital for applying giant magnetostrictive materials in high-power rotary ultrasonic vibration. To further improve the prediction accuracy of the models under high magnetic fields and high prestress, Zheng [13] used hyperbolic tangent functions to calculate the nonlinear strain associated with prestressing based on the physical phenomena related to the rotation of magnetic domains. Subsequently, Jin [14] incorporated temperature effects to develop a coupled magnetic-thermal-force model that considers temperature coupling results. Additionally, this model is also linearized for strains due to magnetic domain rotation under stress. The prevalent phenomenology-based macroscopic principal constitutive model, widely used in engineering applications, cannot address the nature of the nonlinear phenomena caused by magnetic domain rotation and has no apparent physical meaning. Hence, the results are not entirely satisfactory.

From a microscopic perspective, the properties of magnetostrictive materials can be utilized in their polycrystalline state. Based on the quantum mechanics of spin–orbit coupling and magnetic domain wall evolution, the theories provide more physical insights into the microscopic scale of motion for magnetic materials. Despite its better physical interpretation and precise modeling accuracy, it is often not applicable for engineering applications describing macroscopic material objects due to its complexity and high computational requirements. Armstrong [15] first proposed the micro-mechanical modeling of magnetic domain switching of single crystals at the grain scale, which considers different transition strains and spontaneous magnetization. Subsequently, the model is extended to include hysteresis by considering the energy dissipation during domain switching [16]. The micro-mechanical model was recently developed by Hubert [17], considering second-order stress effects, which extended microscale modeling from single crystals to polycrystalline polymers. Yu and Kang [8] proposed a magneto-mechanical, fully coupled, self-consistent homogenization scheme considering elastic-inelastic magneto-static interactions between grains for Terfenol-D. Daniel [18] obtained the strain behavior of Terfenol-D by using the analytical model of the full multiscale method and further verified the analytical model by experiments.

The above article focuses on a single-scale model from macroscopic phenomenology or micro-mechanics, which considers the effect of magnetic domain switching. However, modeling the model from a multiscale perspective can significantly improve the accuracy of modeling and the intrinsic physical meaning representation. As shown in Figure 1, Terfenol-D rods are considered a multilevel structure of magnetic domain-grains-crystals at the microscopic scale. It is possible to derive the mean fields at various scales by linking single-crystal and magnetic domain scales through Boltzmann functions, followed by connecting polycrystalline and single-crystal scales via a self-consistent homogenization scheme. The volumetric mechanical deformation due to different magnetization states is derived from the average value of the volume. The Taylor series expansion of the Gibbs free energy is used to solve the coupling problem of magnetic-elastic-thermal-mechanical multi-physics fields. Finally, the Jiles–Atherton model is utilized to solve the hysteresis behavior in the macroscopic-related multi-physics field. Thus, based on the microscopic phenomenology, this paper demonstrates a multilevel magnetic domain switching scheme, a fully coupled self-consistent homogenization scheme considering elastic-inelastic hysteresis interactions. The proposed model can provide theoretical guidance for the optimal design of rotating giant-magnetostrictive transducers from microscopic and macroscopic multiscale.

## 2. Theoretical and Model

Stress, magnetic field, and temperature acting on the medium produce additional energy, affecting its magnetostrictive characteristics. Thus, the magneto-elastic response of Terfenol-D can be viewed as a result of a complex interplay between various energetic contributions, including exchange energy *w^ex^*, magneto-crystalline energy *w^k^*, magneto-static energy *w^mag^*, and elastic energy *w^el^* from the perspective of quantum mechanics. The exchange energy *w^ex^* is related to Weiss’ molecular field caused by the coupling effect between neighboring atoms, tending to favor a uniform magnetization by aligning the adjacent atomic magnetic moments parallel to each other. The energy unit volume can be represented by the tensor form [16]
(1)$$w^{ex}=\frac{1}{2}\mu_{o}\eta \delta_{kl}M_{k}M_{l}$$
where $\mu_{o}$ is the vacuum permeability, $\delta_{kl}$ is Kronecker delta, η is the coefficient of Weiss’ molecular field, and *k*, *l* are tensor indices.

The magneto-static energy *w^mag^* tends to favor domains for alignment in the magnetic field direction energetically. When subjected to applied magnetic fields, ferromagnetic materials generate demagnetizing fields due to shape anisotropy and internal defects. In a demagnetizing field, the magneto-static energy per unit volume of a magnetization dipole is called the demagnetization energy density. As a result of this energy, magnetostrictive material tends to reduce its total magnetic moment, which influences the response of the magnetostrictive devices. Defining demagnetization energy density as [19]
(2)$$w^{d}=-\frac{1}{2}\mu_{o}N_{kl}M_{k}M_{l}$$
where $N_{kl}$ is the demagnetizing factor, which comprises the internal demagnetizing factor due to internal defects and the geometric or shape demagnetizing factor resulting from shape anisotropy.

Magneto-crystalline energy *w^k^* aligns magnetization along particular axes, called “easy axes”. For the Terfenol-D, the easy axis is along the <111> orientation, i.e., eight directions. Alternatively, if magneto-crystalline anisotropy energy within a crystal is not uniform, it contributes to the evolution of magnetostriction strains, i.e., *w^k^* is not a constant. Then, the rotation mechanism occurs in the magnetic domains. Magneto-elastic interactions occur in ferromagnetic crystals because of elastic energy *w^el^*, which depends on external stress $\sigma_{ij}$. The two energies can be expressed by domain energy *W^α^*, which can be explained in detail below. In this part, the expressions of a nonlinear constitutive model with hysteresis in giant magnetostrictive materials are given from several physical aspects. Considering the positive interaction between the magnetic moments of domains and the demagnetization effect reducing the total magnetic moment, the internal energy per unit volume of the giant magnetostrictive material is expressed as
(3)$$U^{\prime }=U(\lambda_{ij},M_{k},S)+w^{ex}+w^{d}$$
where $U(\lambda_{ij},M_{k},S)$ is the internal energy density, and $dU=\sigma_{ij}d\lambda_{ij}+\mu_{o}H_{k}dM_{k}+TdS$. *σ_ij_*, *λ_ij_*, *µ_o_*, *H_k_*, *M_k_*, *T*, and *S* are the components of field variables participating in the multi-physics coupling of stress, strain, the permeability of the free space, magnetic field, magnetization, temperature, and entropy, respectively. *µ_o_* = 4π × 10^−7^ H/m is the permeability of the free space.

The corresponding Gibbs free energy density function for the thermodynamic system accounting for magneto-elastic-thermo-mechanical stimulus is defined as
(4)$$G\left(\sigma_{ij},M_{k},T\right)=U^{\prime }-TS-\sigma_{ij}\lambda_{ij}$$

The corresponding thermodynamic relation can be derived in differential form as following
(5)$$\lambda_{ij}=-\frac{\partial G}{\partial \sigma_{ij}},\;\mu_{0}H_{k}=\frac{\partial G}{\partial M_{k}},\;S=-\frac{\partial G}{\partial T}$$

To achieve this polynomial constitutive model, choosing the stress *σ_ij_*, the magnetization *M_k_*, and the temperature *T* as independent variables, and the function *G* can be rewritten by a Taylor expansion around the reference point (*σ_ij_*, *M_k_*, *T_r_*) = (0, 0, 20°). *T_r_* is the spin reorientation temperature of the magnetic domains, and usually *T_r_* = 20 °C for Terfenol-D at room temperature. Then, the magnetostrictive constitutive relations of multi-field coupling can be expressed in the following compact form:(6)$$\lambda_{ij}=\lambda_{ij}^{\left(0\right)}\left(\sigma_{kl}\right)+\lambda_{ij}^{\left(1\right)}\left(\sigma_{kl},T\right)+\lambda_{ij}^{\left(2\right)}\left(M_{k},\sigma_{mn},T\right)+\lambda_{ij}^{\left(3\right)}\left(M_{k},T\right)$$
(7)$$H_{k}=H_{k}^{\left(0\right)}\left(M_{l},T\right)+H_{k}^{\left(1\right)}\left(M_{l},\sigma_{mn},T\right)+M_{k}(N_{mk}-\eta \delta_{mk})$$
where Δ*T* is the temperature difference from the room temperature. Equations (6) and (7) take the correlation of material-related parameters as the starting point to deduce the constitutive model of multi-physics coupling. Obviously, before determining the stress *σ_ij_* and magnetization *M*, Equations (6) and (7) are not the final expressions of the constitutive relationship. Compared with the Zheng-Liu model, high-order temperature-dependent dependencies can achieve higher model accuracy prediction. Still, it also leads to multi-material parameter shaping and more complex mathematical polynomial expression dependence. The modified model in this paper is dedicated to meeting the actual needs of engineering applications through less material parameter re-plasticity while considering good accuracy predictions.

### 2.1. On $\lambda_{ij}^{\left(0\right)}\left(\sigma_{kl}\right)$

The rotation of the magnetic moment within a magnetic domain is generally considered the primary mechanism for changes in elastic modulus observed in giant magnetostrictive materials [20,21]. This phenomenon occurs due to spontaneous lattice deformation within each magnetic domain, which is influenced by spin–orbit and orbital-lattice coupling aligned with the magnetization direction. The strain axis rotates with the rotation of the domain magnetization, resulting in the overall deformation of the material. The conditions causing torque rotation include the change of magnetic boundary conditions, the change of AC and DC magnetic fields, and the change in stress amplitude. Usually, the evolution of the elastic modulus of magnetostrictive materials caused by torque rotation giving rise to the shift in stress is considered a magneto-elastic effect. In contrast, the change in the elastic modulus of magnetostrictive materials caused by the reorientation of the domain generated by the magnetic field is considered a magnetostrictive strain. Figure 1 schematically illustrates the magneto-elastic and magnetostrictive effects in a planar magnetostrictive material composed of grain couples, which exhibits a magneto-crystalline anisotropy favoring domain alignment along two mutually orthogonal directions. An ellipse and arrow represent the nonspherical electron cloud and the orientation of magnetic moment, respectively, in the case of the simplified model. Deformation depending on magneto-elastic and magnetostrictive effects due to interaction between the force and magnetization can be described in detail. Based on the small perturbation hypothesis of the magneto-elastic coupling, the dependence of nonspherical electricity on the reorientation of the magnetic domain can be expressed by the rigid connection. The ideal crystal orientation of each grain coincides with the circular specimen’s symmetry axes. In practice, in the case of the initial state, crystals are usually slightly misaligned from the direction of the ideal orientation, as shown in Figure 1(a). When crystals are applied by minor compressive stress (b), domains rotate along the direction away from the compression axis and towards a plane perpendicular to the other direction of the orthogonal; this reorientation attributable to the rotation mechanism continues before the level of compression force increases to a significant level of magnitude (c), where the domains jump to approximately the easy directions closest to the direction of the perpendicular plane under a saturation compression force *σ_s_* (d), with all domains accomplishing rearrangement in the vertical plane (i.e., mechanical saturation); however, in the case of the compression force from (b) to (d), the magnetization of the magnetostrictive material always remains zero. Similarly to the domain above rotation generated by compression force, when crystals are applied by a small magnetic field (e), domains rotate along the direction of the field. Rotation towards the field direction continues as the field increases (f). When the field is equal to (g), the anisotropies of the magneto-crystalline induced by compression forces are overcome, and all domains align along the field direction (i.e., magnetic saturation). It is worth noting that in the field-induced field from (e) to (g), the compression force constantly maintains *σ_s_*. The difference in strain from (d) to (g) induced by the field caused by a 90-degree rotation of the magnetic domain denotes saturation of the magnetostrictive strain *λ_s_*. Therefore, increasing the maximum magnetostrictive strain *λ_m_* under a given pre-press equals the pre-compression attributed to the domain mechanism. Finally, the total elastic strain in the prestressing stage can be divided into linear and nonlinear parts. The former is related to the intrinsic Young’s modulus of the magnetostrictive material, which can be expressed by Hooke’s law by introducing a fourth-order compliance tensor *S_ijk_*; the lateral is related to the spontaneous rotation of magnetic domains, which can be represented by the nonlinear function *λ_ij_* (*σ*) at the microscopic scale. Subsequently, the elastic strain dependent on stress in Equation (6) can be expressed as
(8)$$\lambda_{ij}^{\left(0\right)}\left(\sigma_{kl}\right)=S_{ijkl}\sigma_{kl}+\lambda_{0ij}\left(\sigma_{mn}\right)$$

The linearly elastic strain related to the flexibility tensor *S_ijk_* can be written as [14]:(9)$$S_{ijkl}\sigma_{kl}=\frac{1}{E}\left[\left(1+v\right)\sigma_{ij}-\nu \sigma_{kk}\delta_{ij}\right]$$

*E* and *v* represent Young’s modulus of material at saturation and Poisson’s ratio, respectively.

Let us consider a matter point x on a macroscopic scale in GMM with the cubic crystal structure (see Figure 1) in the magnetostrictive rotary ultrasonic machining composite system (see Figure 1(1)). The volume of the infinite neighborhood of point x located in the region occupied by GMM on such a scale is represented as dv_x_. As shown in Figure 1(2), in the case of the smaller spatial scale (i.e., polycrystalline scale), the microstructure of the grain appears represented by a material point x_1_. This means that the infinitesimal neighborhood dv_x_ in the case of the macroscopic scale becomes a representative volume element (RVE) with aggregated volume v1 at the polycrystalline scale dv_x_ = v_x1_. Let us consider a matter point x_1_ in a polycrystalline scale where the volume of an infinite neighborhood of point x_1_ is represented as dv_x1_. Similarly, at a smaller spatial scale (grain size), microstructures (magnetic domains) appear at such a material point x1. Then, the infinitesimal neighborhood dv_x1_ in the polycrystalline scale becomes a representative volume element with aggregated volume v11 at the grain scale, namely dv_x1_ = v_x11_. Therefore, the multiscale model involves three spatial coordinates x, x1, and x11 of different scales, i.e., macroscale, polycrystalline, and grain scale. From the perspective of the spatial scale of grains and magnetic domains, the single crystal contains multiple magnetic domains. Each magnetic domain has a preferred magnetic orientation direction (easy axis). Since the magnetic domain magnetization can only follow the preferred easy axis direction, the single crystal can be divided into the magnetic domain families (α). Each magnetic domain family is associated with the corresponding easy axis direction. There are only six possible domain families along the axis <100> (100,$\bar{1}00$,010,$0\bar{1}0$,001 and $00\bar{1}$) and eight possible domain families along the axis <111> (111,$\bar{111}$,$\bar{1}11$,$1\bar{11}$,$1\bar{1}1$,$\bar{1}1\bar{1}$,$11\bar{1}$ and $\bar{11}1$). The Boltzmann function can determine the volume fraction of the region α of the magnetic domain family on the grain in the eight <111> directions.
(10)$$f^{\alpha }=\frac{e^{\left(-A_{s}\cdot W^{\alpha }\right)}}{\int_{\alpha }e^{\left(-A_{s}\cdot W^{\alpha }\right)}}$$
where *A_s_* is the material parameter to be identified, connecting the anhysteretic susceptibility and the saturation magnetization, i.e., $A_{s}=\left(3\chi_{m}\right)/\left(\mu_{0}M_{s}^{2}\right)$. The *W^α^* is the domain energy of the famous domain *α* in the grain. The total volume fraction of the α region of the magnetic domain family is 1, i.e., $\int_{\alpha }f^{\alpha }=1$. For cubic magnetostrictive crystals, the deformation related to local magnetization can be described by the following stress-free strain [22].
(11)$$\begin{array}{c} \lambda_{11}^{}=\frac{3}{2}\lambda_{100}\left(\gamma_{1}^{2}-\frac{1}{3}\right),\;\lambda_{12}=\lambda_{111}\sqrt{2}\gamma_{1}\gamma_{2},\;\lambda_{22}=\lambda_{100}\left(\gamma_{2}^{2}-1/3\right)\\ \lambda_{33}=\lambda_{100}\left(\gamma_{3}^{2}-1/3\right),\;\lambda_{13}=\lambda_{111}\sqrt{2}\gamma_{1}\gamma_{3},\;\lambda_{23}=\lambda_{111}\sqrt{2}\gamma_{2}\gamma_{3} \end{array}$$
where *γ* = (cos *θ*sin *Φ*; sin *θ*sin *Φ*; cos *Φ*) is the direction cosines seen as a function of spherical angle *θ* and *Φ* in the range (0–2*π*) and (0–*π*) in the given direction of domain *α*. *λ*_100_ and *λ*_111_ are magnetostrictive constants along <100> and <111> directions, respectively. The interaction between mechanical stress and crystal structure affects the change of easy magnetization direction. In cubic crystallographic symmetry, the stress-induced anisotropy energy density $W_{\alpha }$ depends on the magnetostriction constants *λ*_100_ and *λ*_111_. Assume that both *λ*_100_ and *λ*_111_ are equal to the saturation magnetostrictive constant $\lambda_{s}$, defined as the strain change resulting from the rotation of an utterly random magnetization state in the <100> or <111> direction to a fully oriented magnetization state, respectively.
(12)$$W_{\alpha }^{}=-\sigma_{ij}:\left(\lambda_{0ij}\right)$$

Then, the stress-induced anisotropic density $W_{\alpha }^{}$ in the <111> easy magnetization direction can be calculated by
(13)$$W_{\alpha }=\left\{ \begin{array}{l} W_{111}=W_{\bar{111}}=-\lambda_{s}\left[\sigma_{12}+\sigma_{13}+\sigma_{23}\right]\;\\ W_{\bar{1}11}=W_{\bar{11}1}=-\lambda_{s}\left[-\sigma_{12}-\sigma_{13}+\sigma_{23}\right]\\ W_{1\bar{1}1}=W_{\bar{1}1\bar{1}}=-\lambda_{s}\left[-\sigma_{12}+\sigma_{13}-\sigma_{23}\right]\\ W_{11\bar{1}}=W_{\bar{11}1}=-\lambda_{s}\left[\sigma_{12}-\sigma_{13}-\sigma_{23}\right] \end{array}\right.$$

By merging Formulas (10) and (13), each domain’s explicit expression of volume fraction can be reformulated.
(14)$$f^{\alpha }=\left\{ \begin{array}{l} f^{{}_{111}}=f^{{}_{\bar{111}}}=e^{A_{S}\lambda_{s}(\sigma_{12}+\sigma_{13}+\sigma_{23})}/2\sum \\ f^{{}_{\bar{1}11}}=f^{{}_{\bar{11}1}}=e^{A_{S}\lambda_{s}(-\sigma_{12}-\sigma_{13}+\sigma_{23})}/2\sum \\ f^{{}_{1\bar{1}1}}=f^{{}_{\bar{1}1\bar{1}}}=e^{A_{S}\lambda_{s}(-\sigma_{12}+\sigma_{13}-\sigma_{23})}/2\sum \\ f^{{}_{11\bar{1}}}=f^{{}_{\bar{11}1}}=e^{A_{S}\lambda_{s}(\sigma_{12}-\sigma_{13}-\sigma_{23})}/2\sum \end{array}\right.$$
where $\sum =\sum_{1}^{3}e^{A_{S}\lambda_{s}\sigma_{ij}/3}$.

Then, the magnetostrictive strain in each domain can be expressed as the following form:(15)$$\lambda^{\alpha }=\lambda_{s}\left\{ \begin{array}{l} \lambda^{{}_{111}}=\lambda^{{}_{\bar{111}}}=\{1\;1\;1\;0\;0\;0\;0\;0\}\\ \lambda^{{}_{\bar{1}11}}=\lambda^{{}_{\bar{11}1}}=\{-1\;-1\;1\;0\;0\;0\;0\;0\}\\ \lambda^{{}_{1\bar{1}1}}=\lambda^{{}_{\bar{1}1\bar{1}}}=\{-1\;1\;-1\;0\;0\;0\;0\;0\}\\ \lambda^{{}_{11\bar{1}}}=\lambda^{{}_{\bar{11}1}}=\{1\;-1\;-1\;0\;0\;0\;0\;0\} \end{array}\right.$$

The macroscopic magnetostrictive strain is obtained by averaging the volume of each magnetic domain under arbitrary force.
(16)$$\lambda_{0}={\sum_{1}^{8}\lambda }^{\alpha }f^{\alpha }$$

Finally, the nonlinear elastic strain related to prestress can be obtained:(17)$$\lambda_{0ij}=\lambda_{s}(4e^{{}^{A_{{}_{s}}\lambda_{s}\sigma_{ij}/3}}/3-1)/\sum$$

### 2.2. On $\lambda_{ij}^{\left(1\right)}\left(\sigma_{kl},T\right)$

The temperature-dependent strain $\lambda_{ij}^{1}\left(\sigma_{kl},T\right)$ depends on the thermal volume expansion of magnetostrictive material and is directly related to temperature changes. We can account for it by introducing an isotropic coefficient of thermal expansion $b_{ij}$, that is,
(18)$$\lambda_{ij}^{\left(1\right)}\left(\sigma_{kl},T\right)=b_{ij}\Delta T$$

### 2.3. On $\lambda_{ij}^{\left(2\right)}\left(M_{k},\sigma_{mn},T\right)$

The terms $\lambda_{ij}^{\left(2\right)}\left(M_{k},\sigma_{mn},T\right)$ in Equation (6) dependent on magneto-thermo-elastic coupled response represent the nonlinear magnetostrictive strain. The level of magnetostrictive strain in the free stress condition is zero before the magnetostrictive material is magnetized. In contrast, it approaches a saturation strain value *λ_s_* in the case of saturated magnetization $M_{s}^{T}$. Based on the magneto-elastic effect, the saturated strain value *λ_s_* can be described by introducing a fourth-order tensor *m_ijkl_* on the magnetization *M*. It can be seen that the enhancement of the maximum magnetostrictive strain *λ_max_* under a given pre-press condition is just equal to the pre-compression nonlinear strain attributed to the domain rotation mechanism, i.e., the maximum variation of magnetostrictive strain due to the magneto-elastic coupling effect should be −*λ*_0*ij*_(*σ_mn_*). Then, the terms $\lambda_{ij}^{\left(2\right)}\left(M_{k},\sigma_{mn},T\right)$ related to magnetization *M* and mechanical stress *σ* can be expressed as [23]
(19)$$\begin{array}{l} \lambda_{ij}^{\left(2\right)}\left(M_{k},\sigma_{mn},T\right)\\ =-\frac{1}{2}\left(\begin{array}{l} \frac{\partial^{3}G}{\partial \sigma_{ij}\partial M_{k}\partial M_{l}}+\frac{1}{2}\frac{\partial^{5}G}{\partial \sigma_{ij}\partial M_{k}\partial M_{l}\partial T^{2}}\Delta T^{2}+\cdots \\ +\frac{\partial^{4}G}{\partial \sigma_{ij}\partial \sigma_{mn}\partial M_{k}\partial M_{l}}\sigma_{mn}+\frac{\partial^{5}G}{\partial \sigma_{ij}\partial \sigma_{mn}\partial M_{k}\partial M_{l}\partial T}\sigma_{mn}\Delta T\cdots \end{array}\right)M_{k}M_{l}\\ -\frac{1}{4!}(\frac{\partial^{5}G}{\partial \sigma_{ij}\partial M_{k}\partial M_{l}\partial M_{i}\partial M_{j}}+\frac{\partial^{6}G}{\partial \sigma_{ij}\partial \sigma_{mn}\partial M_{k}\partial M_{l}\partial M_{i}\partial M_{j}}\sigma_{mn}+\cdots )M_{k}M_{l}M_{i}M_{j}\\ =\left[m_{ijkl}-\frac{\int_{\alpha }f^{\alpha }\lambda^{\alpha }}{{\left(M_{s}^{T}\right)}^{2}}\delta_{kl}\right]M_{k}M_{l} \end{array}$$
where $M_{s}^{T}$ is the temperature-dependent saturation magnetization in easy-axis transformation temperature *T_r_* which can be evaluated by the saturation magnetization $M_{s}$ and temperature difference ΔT [24].
(20)$$M_{s}^{T}(\Delta T)=M_{s}{\left(1-\frac{\Delta T}{T_{c}-T_{r}}\right)}^{x}$$
where *T_c_* is the Curie temperature, and *x* is the amplification factor depending on the slope of the saturation magnetization with the temperature.

### 2.4. On $\lambda_{ij}^{\left(3\right)}\left(M_{k},T\right)$

The term $\lambda_{ij}^{\left(3\right)}\left(M_{k},T\right)$ is strain-dependent on the thermal-magneto coupling. Meanwhile, the experimental result also indicates that the saturated magnetostrictive strain of material linearly decreases with the increasing temperature when the temperature is above the easy-axis transition temperature [25,26]. Then it can be expressed as
(21)$$\lambda_{ij}^{\left(3\right)}\left(M_{k},T\right)=\frac{\beta_{\mathrm{ij}}}{{\left(M_{s}^{T}\right)}^{2}}M_{k}M_{l}\Delta T\delta_{kl}$$
where $\beta_{\mathrm{ij}}$ is the slope where temperature deviation is the function of magnetostriction at the saturated magnetization.

### 2.5. On $H_{k}^{\left(0\right)}\left(M_{l},T\right)$

The term $H_{k}^{\left(0\right)}\left(M_{l},T\right)$ correlated to the interaction between magnetization and thermo-magnetic coupling is represented as magnetization *M^T^* = *M* (*H*, *T*) independent of stress and can reach saturation. It can be characterized by a nonlinear function *M^T^* = $M_{S}^{T}$
*f*(*k^T^**H_k_*). Then, $H_{k}^{\left(0\right)}\left(M_{l},T\right)$ related to magnetization can be expressed linearly:(22)$$H_{k}^{\left(0\right)}\left(M_{l},T\right)=f_{k}^{-1}\left(M_{l},\Delta T\right)$$
where the superscript −1 is the inverse operation in the nonlinear function *M^T^* = $M_{S}^{T}$
*f*(*k^T^H*). Considering higher calculation accuracy and better phenomenological physical significance, the Langevin function *f*(*x*) = coth(*x*) − 1/*x* is used due to its better physical background derived from Boltzmann statistics. The response curve of magnetization concerning magnetic in free stress state can be described by a vectorial version of the Langevin function considering an isotropic material relaxing fact *k*. That is,
(23)$$M=M_{x}i+M_{y}j+M_{z}k=M_{s}^{T}f(kH)$$

For instance, the *x* component of the Langevin function can be written as
(24)$$M_{x}=M_{s}^{T}\left[\coth(k^{T}|H|)-\frac{1}{k^{T}|H|}\right]\frac{H_{x}}{\left|H\right|}$$
where $k^{T}=(\pi /2)\chi_{m}/M_{s}^{T}$ is a temperature-dependent relaxation factor and $\chi_{m}$ is the susceptibility of the initial linear region (see [25,26]).

Then, the inversion of the nonlinear isotropic function in Equation (24) can be written as
(25)$$f_{k}^{-1}\left(M_{l},\Delta T\right)=\frac{1}{k^{T}}f^{-1}\left(\frac{M_{l}}{M_{s}^{T}}\right)\delta_{kl}$$

### 2.6. On $H_{k}^{\left(1\right)}\left(M_{l},\sigma_{mn},T\right)$

The coupled term $H_{k}^{\left(1\right)}\left(M_{l},\sigma_{mn},T\right)$ shows the inverse magnetostrictive effect due to the prestress on the magnetostrictive material with ambient temperature and the inverse magneto-thermo-elastic coupling effect, which can be expressed analogous to $\lambda_{ij}^{\left(2\right)}\left(M_{k},\sigma_{mn},T\right)$ and $\lambda_{ij}^{\left(3\right)}\left(M_{k},T\right)$.
(26)$$\begin{array}{l} H_{k}^{\left(1\right)}\left(M_{l},\sigma_{mn},T\right)\\ =\left(\begin{array}{l} \frac{\partial^{3}G}{\partial \sigma_{ij}\partial M_{k}\partial M_{l}}\sigma_{ij}+\frac{1}{2}\frac{\partial^{5}G}{\partial \sigma_{ij}\partial M_{k}\partial M_{l}\partial T^{2}}\sigma_{ij}\Delta T^{2}\\ +\cdots \frac{1}{2}\frac{\partial^{4}G}{\partial \sigma_{ij}\partial \sigma_{mn}\partial M_{k}\partial M_{l}}\sigma_{ij}\sigma_{mn}+\frac{1}{2}\frac{\partial^{5}G}{\partial \sigma_{ij}\partial \sigma_{mn}\partial M_{k}\partial M_{l}\partial T}\sigma_{ij}\sigma_{mn}\Delta T\cdots \end{array}\right)M_{l}+\frac{\partial^{4}G}{\partial T\partial \sigma_{ij}\partial M_{k}\partial M_{l}}\Delta T\sigma_{ij}M_{l}\\ =-\frac{1}{\mu_{o}}\left[\frac{\lambda_{0ij}^{\left(2\right)}\left(M_{k},\sigma_{mn},T\right)}{M_{k}^{2}}\sigma_{ij}+m_{ijkl}M_{l}M_{k}\sigma_{ij}\right]M_{k}-\frac{2}{\mu_{o}}(\frac{\lambda_{0ij}^{\left(3\right)}\left(M_{k},T\right)}{M_{k}^{}}\sigma_{ij})\\ =-\frac{1}{\mu_{o}}\left[2m_{ijkl}-\frac{\int_{\alpha }f^{\alpha }\varepsilon_{\alpha }}{{\left(M_{s}^{T}\right)}^{2}}\delta_{kl}\right]\sigma_{ij}M_{l}-\frac{2\beta_{ij}\sigma_{ij}M_{l}}{\mu_{o}{\left(M_{s}^{T}\right)}^{2}}\Delta T\delta_{kl} \end{array}$$

In the isotropic case, taking account of the symmetry of the magnetostrictive tensor $m_{ijkl}$, it can be represented in terms of saturation strain and temperature-dependent saturation magnetization as follows:(27)$$m_{ijkl}=\frac{\lambda_{s}}{{\left(M_{s}^{T}\right)}^{2}}\left[\frac{3}{4}\left(\delta_{ik}\delta_{jl}+\delta_{il}\delta_{jk}\right)-\frac{1}{2}\delta_{ij}\delta_{kl}\right]$$

Thus, one can obtain
(28)$$m_{ijkl}M_{k}M_{l}\sigma_{ij}=\frac{\lambda_{s}\sigma_{ij}}{{\left(M_{s}^{T}\right)}^{2}}\left[\frac{3}{2}M_{i}M_{j}-\frac{1}{2}M_{k}M_{k}\delta_{ij}\right]$$

Based on the above expressions, we obtain
(29)$$m_{ijkl}\sigma_{ij}=\frac{\lambda_{s}}{{\left(M_{s}^{T}\right)}^{2}}\left[\frac{3}{2}\sigma_{ij}-\frac{1}{2}\sigma_{ij}\delta_{ij}\right]=\frac{\lambda_{s}}{{\left(M_{s}^{T}\right)}^{2}}{\tilde{\sigma }}_{ij}\delta_{ij}$$
where ${\tilde{\sigma }}_{ij}=3\sigma_{ij}/2-\sigma_{ij}\delta_{ij}/2$ is 3/2 times as much as the stress deviation part of the stress tensor $\sigma_{ij}$ and ${\tilde{\sigma }}_{ij}\delta_{ij}={\tilde{\sigma }}_{kk}=0$ in the volume magnetostrictive material.

In practical applications, magnetostrictive materials are typically used as prestressed rods or thin films subjected to magnetic fields; the magneto-thermo-elastic coupling effect induces magnetization and outputs magnetostriction. According to Equations (8), (17)–(19), (21) and (26), in the case of an isotropic magnetostrictive material, a new pair of a magneto-elastic-thermo-mechanical coupled constitutive model for magnetostrictive material can be expressed in the following compact form in the 3-D case:(30)$$\begin{array}{l} \lambda_{ij}=\frac{1}{E_{in}}\left[(1+v)\sigma_{ij}-v\sigma_{kk}\delta_{ij}\right]+\frac{\lambda_{s}}{{\left(M_{s}^{T}\right)}^{2}}\left[\frac{3}{2}M_{i}M_{j}-\frac{1}{2}M_{k}M_{l}\delta_{ij}\right]\\ \;\;\;\;\;+\Delta T\left[b_{ij}+\frac{\beta_{ij}\delta_{kl}}{{\left(M_{s}^{T}\right)}^{2}}M_{k}M_{l}\right]+\frac{\lambda_{s}4(e^{{}^{A_{{}_{s}}\lambda_{s}\sigma_{ij}/3}}-1)}{3\sum }\delta_{kl}(1-\frac{M_{k}M_{k}}{{\left(M_{s}^{T}\right)}^{2}}) \end{array}$$
(31)$$\begin{array}{l} H_{k}=\frac{1}{k}f^{-1}\left(\frac{M_{l}}{M_{s}^{T}}\right)\delta_{kl}+M_{l}\left[N_{kl}-\left(\eta +\frac{2\beta_{{}_{ij}}\sigma_{ij}\Delta T}{\mu_{o}{\left(M_{s}^{T}\right)}^{2}}\right)\delta_{kl}\right]\\ \;\;\;\;\;-\frac{M_{l}}{\mu_{o}{\left(M_{s}^{T}\right)}^{2}}\left(2{\tilde{\sigma }}_{kl}\lambda_{s}-\frac{\lambda_{s}4(e^{{}^{A_{{}_{s}}\lambda_{s}\sigma_{ij}/3}}-1)}{3\sum }\delta_{kl}\right) \end{array}$$

Although microscopically based phenomenological formulation has a clear and hierarchical physical explanation, it cannot be utilized to account for the hysteresis behavior of magnetostrictive material due to supposing that magnetization is equal to anhysteretic magnetization *M_an_*.

## 3. Hysteresis Effect

In this section, based on the improved vector Jiles–Atherton hysteresis model [27], a magneto-elastic-thermo-mechanical coupled constitutive model with hysteresis is obtained. In the J–A hysteresis model, the total magnetization *M* is made of two parcels: the reversible magnetization *M_rev_* representing the moment rotation, and the irreversible magnetization *M_irr_* illustrating the obstruction mechanism of the domain wall movement caused by the domain wall pinning. Then, the total magnetization *M* must follow the anhysteretic magnetization *M_an_* when ignoring the energy losses. To quantify the *M_an_* of a giant magnetostrictive material, it is necessary first to determine the effective magnetic field that acts inside the material.

According to the above discussion, in addition to the Weiss molecular field and the demagnetization field, the coupling effect of stress and temperature on the magnetostrictive strain also changes the effective magnetic field *H_e_* inside the giant magnetostrictive material, resulting in a change in magnetization. The effective magnetic field *H_e_* is directly related to the ideal magnetization *M_an_* and can be expressed as
(32)$$H_{i}^{e}=H_{i}+\alpha_{ij}M_{i}^{\mathrm{an}}$$
where $\alpha_{ij}$ can be determined by rearranging Equation (32):(33)$$\alpha_{ij}=M_{l}\left[N_{kl}-\left(\eta +\frac{2\beta_{{}_{ij}}\sigma_{ij}\Delta T}{\mu_{o}{\left(M_{s}^{T}\right)}^{2}}\right)\delta_{kl}\right]-\frac{M_{l}}{\mu_{o}{\left(M_{s}^{T}\right)}^{2}}\left(2{\tilde{\sigma }}_{kl}\lambda_{s}-\frac{\lambda_{s}4(e^{{}^{A_{{}_{s}}\lambda_{s}\sigma_{ij}/3}}-1)}{3\sum }\delta_{kl}\right)$$

Based on Boltzmann statistics, the ideal magnetization *M_an_* in Equation (33) is approximated by the Langevin function as mentioned previously in Equation (25). It can be expressed in vector form as
(34)$$M_{i}^{an}=M_{s}^{T}\left[\coth(k^{T}|H_{i}^{e}|)-\frac{1}{k^{T}|H_{i}^{e}|}\right]\frac{H_{i}^{e}}{\left|H_{i}^{e}\right|}$$

Then, the derivative of $M_{i}^{an}$ to $H_{i}^{e}$ is calculated as follows:(35)$$\frac{\partial M_{i}^{an}}{\partial H_{i}^{e}}=\frac{2}{\pi \chi_{m}}{\left(\frac{M_{s}^{T}}{H_{i}^{e}}\right)}^{2}-\frac{\pi \chi_{m}}{2\sin h^{2}\left(\pi \chi_{m}H_{i}^{e}/2M_{s}^{T}\right)}$$

Hysteresis occurs when the domain wall movements are restricted; this is caused by defects such as residual stress, the doping of non-magnetic particles, and holes generated during material preparation. Considering these imperfections jointly as pinning sites, impedance to motion can be measured by a pinning constant *K*. Leite [28] introduced an auxiliary vector variable $\chi^{\prime }=K_{ij}^{-1}\left(M_{an}-M_{irr}\right)$ related to the pinning constant *K*, which is a second-order symmetric tensor with the same diagonal term in the isotropic case. If the variation $M_{irr}$ is parallel to $\chi^{\prime }$, then an incremental expansion $M_{irr}$ can be obtained as
(36)$$dM_{i}^{irr}=\chi^{\prime }\cdot {\left|\chi^{\prime }\right|}^{-1}{\left[\chi^{\prime }\cdot dH_{i}^{e}\right]}^{+}$$
where the notation [.]^+^ takes the value [.] for [.] > 0 and 0 for [.] ≤ 0.

On the other hand, the reversible magnetization $M_{\mathrm{rev}}$ can be interpreted as the motive force for domain wall movement and is proportional to the difference between $M_{\mathrm{an}}$ and $M_{\mathrm{irr}}$; then, it can be expressed as
(37)$$dM_{i}^{\mathrm{rev}}=c_{ik}\left(dM_{i}^{\mathrm{an}}-dM_{i}^{\mathrm{irr}}\right)$$

Combining Equations (38) and (39) and then utilizing $\chi^{\prime }=K_{ij}^{-1}\left(M_{\mathrm{an}}-M_{\mathrm{irr}}\right)$ yields the overall magnetization *M* in the incremental form as
(38)$$dM=\chi \cdot |\chi {|}^{-1}{\left[\chi \cdot dH_{i}^{e}\right]}^{+}+c\cdot dM_{i}^{\mathrm{an}}$$
where $\chi =K_{ij}\left(M_{\mathrm{ij}}^{\mathrm{an}}-M_{ij}\right)$.

For better numerical simulation, Equation (38) can be expressed in terms of magnetic variable *H* with the aid of Equations (34) and (36).
(39)$$dM=\left\{ \begin{array}{c} \left[\chi |\chi {|}^{-1}\chi +\overset{}{c}\left(\begin{array}{l} \frac{2}{\pi \chi_{m}}{\left(\frac{M_{s}^{T}}{H_{i}^{e}}\right)}^{2}-\\ \frac{\pi \chi_{m}}{2\sin h^{2}\left(\pi \chi_{m}H_{i}^{e}/2M_{s}^{T}\right)} \end{array}\right)\right]dH_{\mathrm{eff}\;}\;\\ \chi \cdot dH_{\mathrm{eff}}>0\\ \overset{}{c}\left(\begin{array}{c} \frac{2}{\pi \chi_{m}}{\left(\frac{M_{s}^{T}}{H_{i}^{e}}\right)}^{2}-\\ \frac{\pi \chi_{m}}{2\sin h^{2}\left(\pi \chi_{m}H_{i}^{e}/2M_{s}^{T}\right)} \end{array}\right)dH_{\mathrm{eff}\;}\;\\ \chi \cdot dH_{\mathrm{eff}}\leq 0 \end{array}\right.$$

Equation (39) establishes the relationship of *H-M* and then can be utilized to acquire the total magnetization hysteresis when the magnetic field is applied in an incremental direction or vice versa. Moreover, the total strain coupling magneto-elastic-thermo-mechanical with hysteresis behavior can be obtained by Equation (30) under the condition that the total hysteresis *M* is determined preferentially. Then, a nonlinear magneto-elastic-thermo-mechanical coupled constitutive model with hysteresis for giant magnetostrictive materials is achieved.

### Model Implementation of Total Magnetization M

Figure 2 is the solution flow chart of total magnetization *M* versus effective magnetic field *H_e_*. The temperature-dependent material parameter *K^T^* and the macroscopic field variable parameters are necessary input data, which can be obtained by experimental optimization in the physical sense. The anhysteresis magnetization $M_{\mathrm{an}}^{i}$ is first calculated by the initial effective magnetic field $H_{e}^{i}$ using Equation (34), and the result is returned to Equation (32) to obtain the updated $H_{e}^{i+1}$. Repeating the initial process, the anhysteresis magnetization $M_{\mathrm{an}}^{i+1}$ is acquired. This process is iterated until the convergence test results meet the requirements, i.e., $\left|M_{\mathrm{an}}^{i+1}-M_{\mathrm{an}}^{i}\right|/\left|M_{\mathrm{an}}^{i}\right|⩽0.01\%$.

The above derivation process shows that the multi-physical field coupling nonlinear transient model obtained has two essential features. First of all, based on a microscopically based multiscale approach, the increment of the maximum magnetostrictive attributable to the pre-contraction as a result of the domain rotation under prestress is calculated, i.e., Equation (16). Thereby, the multi-physical field nonlinear model has the capacity to reveal the variation mechanism of magnetostrictive material from microscopical magneto-elastic-thermo-mechanical couple to phenomenological magnetostrictive strain. Moreover, utilizing the improved vector Jiles–Atherton hysteresis model, this nonlinear constitutive model obtained sufficiently considers the hysteresis behavior of magnetization *M* and magnetostrictive strain λ, which are related to a microscopically magnetic state.

## 4. Model Verification and Discussion

This section discusses the numerical results of the proposed model by examining a case study of a one-dimensional rod and, more importantly, a two-dimensional plate. First, we compare the predicted constitutive behavior with the available experimental data on magnetostriction and magnetization caused by various external stimuli. Next, a parametric study is conducted to reveal the variation pattern in magnetization and magnetostriction curves as a function of the magnetic field and temperature to assess the combined effect of prestressing and temperature. Modified material parameters for the Terfenol-D is shown in Table 1.

We propose a microscopic phenomenology-based physical approach to estimate magnetostrictive strain and magnetization phenomena under different mixed external stimulus conditions, in which magnetostrictive strain and magnetization phenomena depend on the magnetization state of the three-dimensional magnetostrictive material. First, without considering the hysteresis case, Figure 3 depicts the magnetization and magnetostriction curves of the present constitutive model at room temperature and under different prestress conditions. The material parameters are calibrated from the ZL model reported. As shown in Figure 3, the present constitutive model’s predictions generally agree with the ZL model results at varying stresses at lower and higher magnetic fields.

Next, we use the experimental results reported by Moffett et al. [30] to verify the magnetostrictive hysteresis results of the theoretical model for the same external environment (room temperature) and different prestress excitation conditions (−6.9 to −65.4 MPa). Again, the material parameters are chosen from the calibration and optimization in Moffett’s experimental results. A comparison of the predicted and experimental results of magnetostrictive hysteresis of the three-dimensional rod-shaped magnetostrictive material under cyclic magnetic and different compressive stress loading at ambient room temperature is shown in Figure 4. The results show that the magnetostrictive hysteresis results predicted by the proposed theoretical model show good dependence and correlation between external stimuli and experimental results over the entire range of magnetically coupled fields. Additionally, these predictions by the proposed model are consistent with the experimental results in Figure 3 and Figure 4. Such consistency ensures the applicability and validity of the proposed magnetically coupled hysteresis model.

Third, the experimental results of Gao et al. [31] contain the coupling effect of thermal effects on the magnetostrictive material Terfenol-D rods under different prestress conditions. Then, we also investigate the effect of different ambient temperatures and external operating states on the coupled magnetic-elastic-thermal-mechanical hysteresis effect of three-dimensional magnetostrictive materials. We extract and optimize the experimental material parameters of Gao et al. for the new constitutive model prediction. Figure 5 shows the magnetization and magnetostrictive hysteresis responses of Terfenol-D at varying ambient temperatures and stress states. Figure 5a,b show the magnetization and magnetostrictive hysteresis curves generated by the theoretical model for different compressive prestress at low temperatures of 60 °C and high temperatures of 140 °C as the magnetic field approaches saturation. A significant hysteresis is observed at lower temperatures, especially for more considerable compressive stresses. In addition, the rate of change of magnetization M for the magnetic field H, i.e., the piezomagnetic coefficient, decreases with increasing applied compressive stress in the lower and moderate range of applied axial magnetic fields. The phenomenon can be explained by the fact that, as shown in Figure 1, the applied external compressive stress is more likely to align the magnetic domains in the plane perpendicular to an axial force from the point of view of the domain rotation of the magneto-elastic-mechanical coupling. The external compressive stress acts as an additional mechanical obstacle, making the magnetic moment’s rotational reluctance in otherwise identical conditions more significant and making the magnetization process more difficult. Therefore, the magnetic saturation of the material requires a sufficiently large applied magnetic energy to rotate the magnetic domains.

Similarly, the rate of change of the magnetostriction coefficient λ for the magnetic field H (i.e., dλ/dH) becomes smaller with increasing applied mechanical compressive load in the lower and moderate axial magnetic field range. However, this is not the case for saturation magnetostriction. In the high magnetic field application range, the exact opposite magnetostriction flip phenomenon is exhibited, i.e., the saturation magnetostriction coefficient is proportional to the mechanical load of the applied compressive stress. Similarly, from the angle of domain rotation of the magneto-elastic-thermal-mechanical coupling, which is due to the pre-compression of the domain rotation caused by the mechanical compressive stress before the application of the magnetic field, the higher the mechanical compressive stress, the higher the pre-compression due to the domain rotation, and thus the higher the saturation magnetostriction coefficient induced at the saturation magnetic field. Conversely, it exhibits the opposite property when subjected to tension. For the design of rotating giant magnetostrictive transducers, it is necessary to consider the appropriate pre-compression of the giant magnetostrictive material used for the ultrasonic oscillator to obtain greater ultrasonic vibration energy. On the other hand, the thermal effect in the magneto-elastic-thermal-mechanical coupling becomes evident at a high temperature of 140 °C. The curve is similarly scaled in proportion due to the thermal motion of molecules in the high-temperature environment, which partially disrupts the directional arrangement of magnetic molecular flow.

Fourth, due to the nonlinear nature of Terfenol-D under multi-field coupling conditions, we investigate the variation characteristics of Young’s modulus with magnetic field H under different external environments of force and temperature. Figure 6a depicts the response of Young’s modulus of Terfenol-D at a low temperature of 60° and with the stimuli of various external forces (−25, −35, −50, and −75 Mpa) due to the applied magnetic field. Both mechanical and magnetic loads are along the axial direction. It can be noted that with increasing compressive stress, Young’s modulus decreases and then increases with different magnetic fields and exhibits higher nonlinear characteristics due to the magneto-elastic effect dominating the stress–strain response. The Young’s modulus can approach the intrinsic Young’s modulus E_s_ under more considerable compressive stresses and weaker magnetic fields or more minor compressive stresses and larger magnetic fields due to the combined effect of pure mechanical elasticity and the magneto-elastic coupling of the magnetic domain rotation, which ultimately aligns the domains along the axial direction parallel to the magnetic field or perpendicular to the plane of stress. Figure 6b depicts the temperature response sensitivity of Young’s modulus at −25 MPa and different temperatures of stimulation: 20, 40, 60, and 100 °C. The results show the temperature insensitivity of Young’s modulus at weak magnetic fields, but at strong magnetic fields, Young’s modulus increases with temperature, exhibiting a stable temperature nonlinear property. From the point of view of qualitative analysis, the predicted results are consistent with the experimental results of Kellogg [32] and Zheng [33] regarding the effects of prestressing and temperature on Young’s modulus. To reduce the severe eddy current loss on the surface of the giant magnetostrictive material under high-frequency excitation in the application of ultrasonic vibration machining, as shown in Figure 7, the material can be cut in different ways. The red part is filled with epoxy resin after cutting to increase strength.

Finally, for the practical use of Terfenol-D in rotating giant magnetostrictive ultrasonic transducers where slicing is often used to reduce eddy current, the proposed model is used to simulate the experimental results of Terfenol-D amorphous films under the high prestress reported by Schatz et al. [29] to investigate the response of two-dimensional Terfenol-D under magnetic-elastic-mechanical coupling behavior. Figure 8 shows that the magnetic anisotropy induced by tensile and compressive stresses in the two-dimensional plane exhibits opposite properties, i.e., the easy direction of magnetization under tensile stress is parallel to the in-plane direction. In contrast, the easy direction of magnetization under compressive stress is perpendicular to the in-plane direction. This is because the diaphragm with tensile stress tends to rotate the magnetic domains in the magnetization direction, while a diaphragm with compressive stress tends to rotate the domains in the stress direction, which is consistent with the domain analysis in Figure 1. The rational application of stresses in different directions can change the susceptibility of the magnetic domains to magnetization. This can also provide theoretical design guidance for two-dimensional coupled vibrations of giant magnetostrictive materials in rotational ultrasonic vibrations, such as longitudinal-bending coupled and longitudinal-torsional coupled elliptical ultrasonic vibration processing. We conclude that the microscopic model we established can solve the hysteresis effect of the magneto-elastic-thermal-mechanical coupling of giant magneto-elastic effect materials and provide theoretical guidance for applying giant magnetostrictive materials in rotational ultrasonic vibration processing.

In the practical application of giant magnetostrictive materials in rotary ultrasonic machining, due to the unique manufacturing process, defects such as stress concentration, regional cracks, voids, or weak links will inevitably cause the existence of material demagnetization sources. When the magnetization vector divergence of the material surface changes, the magnetization distribution in the magnetic domain region is no longer uniform, which causes the prediction result of the giant magnetostrictive rotary ultrasonic machining system to be unstable. Figure 8 gives the magneto-elastic coupling prediction results of the two-dimensional giant magnetostrictive sheet material structure, with consideration of the demagnetization factor (N = 1) and without consideration of the contribution of the demagnetization factor. Figure 9 gives the effect of the demagnetization factor and the Weiss molecular field on the hysteresis loop of the two-dimensional giant magnetostrictive sheet material structure, where the demagnetization factor and the Weiss molecular field effect are η=0.089 and N = 1, respectively. Without considering the demagnetization and Weiss molecular fields, the theoretical prediction results may underestimate the energy loss in the magnetization hysteresis loop. To accurately quantify the output strain of the giant magnetostrictive ultrasonic machining system under static loading, the constitutive model should consider the effect of the demagnetization field and the Weiss molecular field. Incorporating them in the effective magnetic field is essential and strongly supports the demagnetization field and Weiss molecular field theory.

## 5. Conclusions

To provide quantitative modeling of the induced magnetostrictive effects in giant magnetostrictive materials under mixed magnetic-force-thermal excitation, a widely used three-dimensional fully coupled magnetostrictive theoretical model is developed by means of considering the interactions between the elastic–inelastic hysteresis effects at the grain, polycrystalline complex, and macroscopic scales. The physical fields at larger spatial scales are obtained by amplifying the information of the corresponding fields at more minor spatial scales. The main conclusions are as follows.

The proposed model can accurately predict giant magnetostrictive materials’ deformation and hysteresis effects in ultrasonic systems by comprehensively considering parametric information at different scales and the coupled magnetic-elastic-thermal-mechanical impacts in a mixed external environment.The impact of various external forces on the stress-induced magnetic anisotropy properties of the two-dimensional magnetostrictive film is being considered. It can be inferred that meticulous attention must be given to these design factors to facilitate the expansion of magnetostrictive materials in ultrasonic processing systems.

## Figures and Tables

**Figure 1 materials-16-04914-f001:**
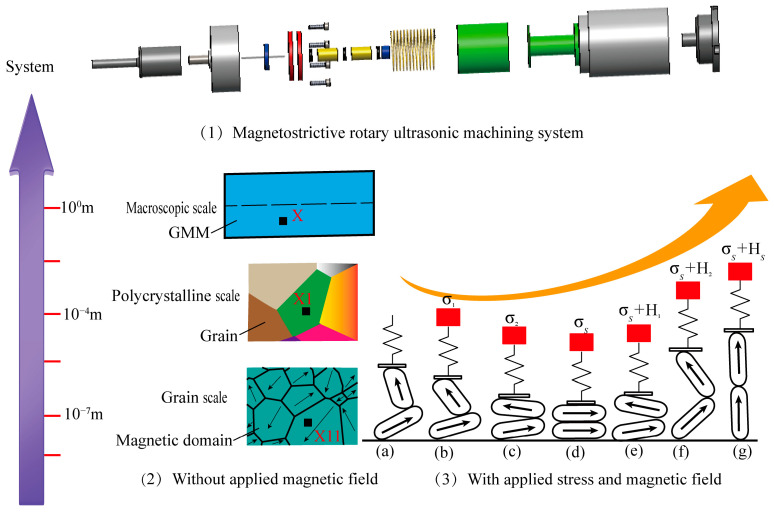
A schematic representation of Terfenol-D from micro to macro scale.

**Figure 2 materials-16-04914-f002:**
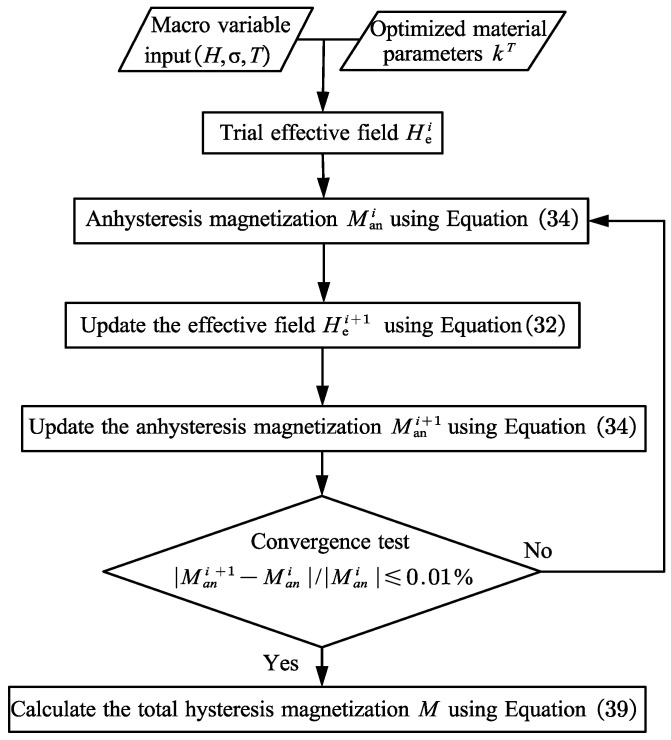
Iterative procedure for the total magnetization *M*.

**Figure 3 materials-16-04914-f003:**
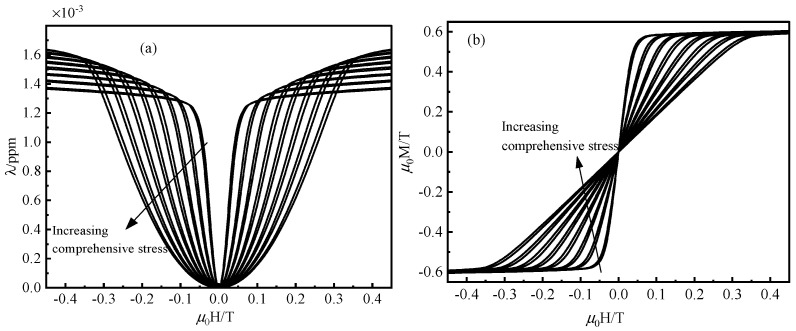
Magnetostrictive (**a**) and magnetization (**b**) curves predicted by the proposed model without considering hysteresis under gradually increasing compressive stress (−6.9 −8.3 −15.3 −17.2 −23.6 −25.5 −32 −34.5 −40.4 −42.8 −48.7 −51.3 −57.1 −59.6 −65.4 and −67.9 MPa) at room temperature.

**Figure 4 materials-16-04914-f004:**
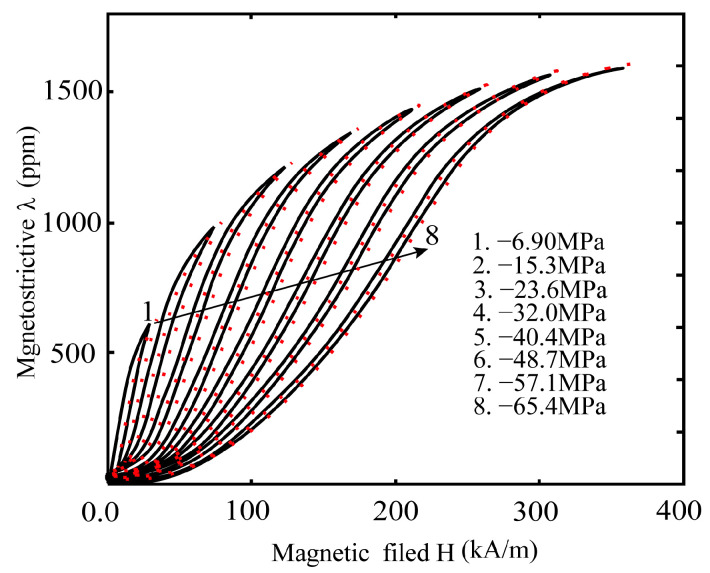
Comparisons of the proposed model to the experiment data (Moffett et al. [30]) on magnetostrictive hysteresis under different prestress at room temperature.

**Figure 5 materials-16-04914-f005:**
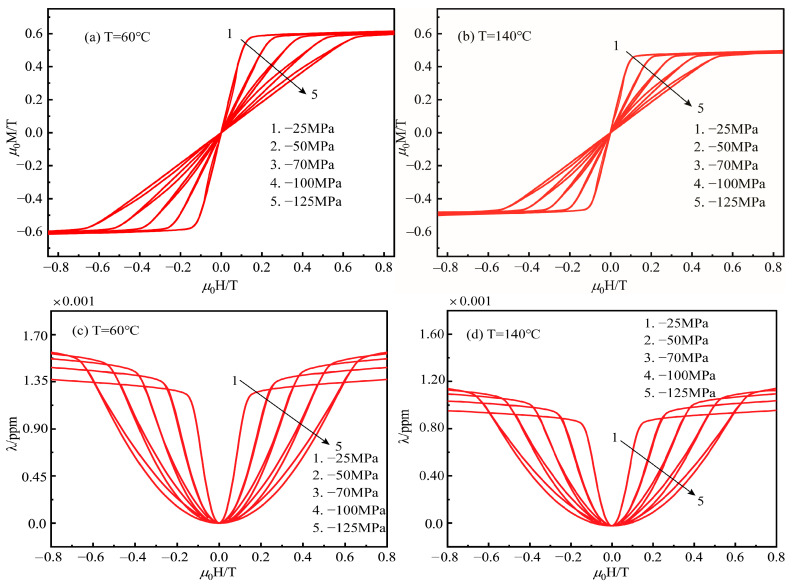
Parametric study of magnetization and magnetostriction hysteresis curves at different forces and temperatures; (**a**–**d**) the relation between M, λ and H for different compressive stress at temperature 60 °C and elevated one 140 °C, respectively.; (**e**,**f**) the relation between λ and H for different tensile at temperature 20 °C and elevated one 120 °C, respectively.

**Figure 6 materials-16-04914-f006:**
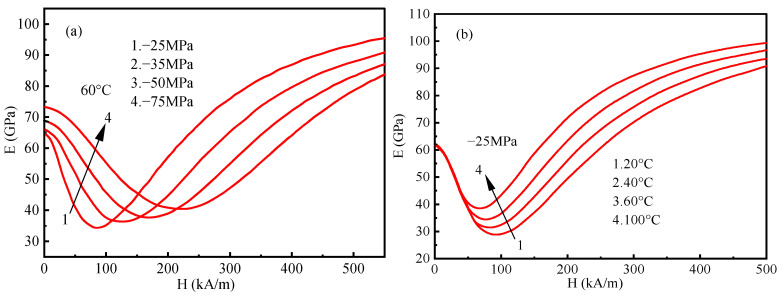
Predicted Young’s modulus vs. magnet filed under different prestress (**a**) and temperature (**b**).

**Figure 7 materials-16-04914-f007:**
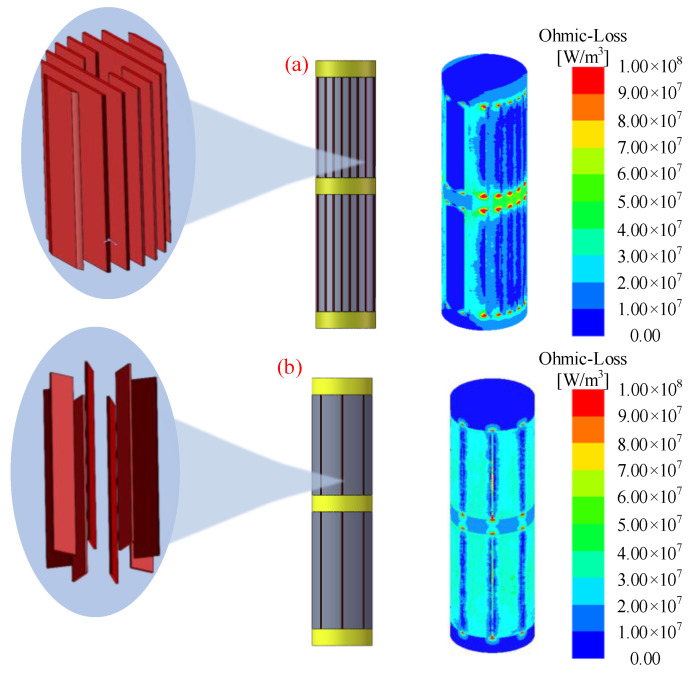
Different treatment methods for reducing eddy current loss; (**a**) slicing; (**b**) cutting.

**Figure 8 materials-16-04914-f008:**
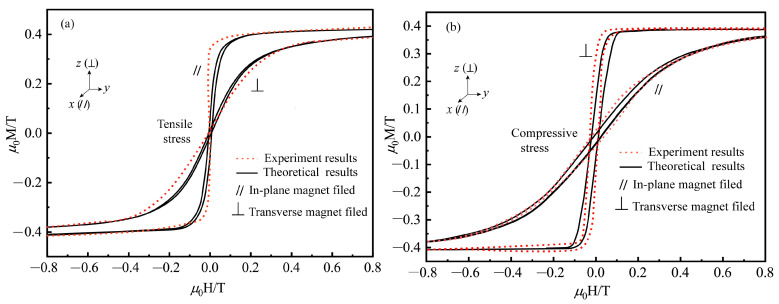
Comparison of the proposed constitutive model with experiment data (Schatz et al. [29]) for a magnetostrictive film on the magnetization hysteresis curve under different field directions and forces; (**a**) tensile stress (**b**) compressive stress.

**Figure 9 materials-16-04914-f009:**
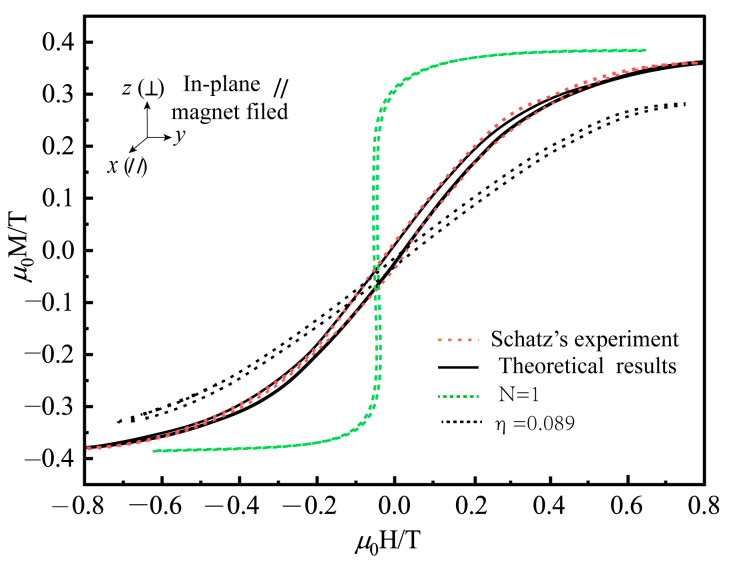
Effect of demagnetization factor and Weiss molecular field on hysteresis curve.

**Table 1 materials-16-04914-t001:** Modified material parameters for the Terfenol-D.

Property	${\left(Tb_{0.27}Dy_{0.73}\right)}_{0.42}Fe_{0.58}$ Schatz et al. [29]	$Tb_{0.30}Dy_{0.70}Fe_{1.93}$ Moffett et al. [30]	$Tb_{0.27}Dy_{0.73}Fe_{1.95}$ Gao et al. [31]
$M_{s}(\mathrm{kA}/m)$	810	660	712
$E_{s}(\mathrm{GPa})$	110	110	110
$\sigma_{s}(\mathrm{MPa})$	60	190	110
$\lambda_{s}(\mathrm{ppm})$	1820	1435	1910
$T_{c}({}^{\circ}C)$	383.3	383.3	383.3
$b(\times 10^{-6}/{}^{\circ}C)$	12	12	12
$\beta (\times 10^{-6}/{}^{\circ}C)$	−3.5	−3.5	−3.5
*C*	0.55	0.81	0.42
$\chi_{m}$	67	+35	24

## Data Availability

Data are contained within the article as figures and tables.

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
