# Peer review of "Domain Switching-Based Nonlinear Coupling Response for Giant Magnetostrictive Materials"

_materials, 2023, doi:10.3390/ma16144914_

Round 1
Reviewer 1 Report
see attachment file

Author Response
We greatly appreciate your guidance and help. We are sincerely thankful for all your comments and suggestions, which have significantly helped us improve the manuscript. This manuscript has been carefully revised based on these valuable suggestions. Additions to the text are shown in yellow. Deletions in the text are not shown.

Reviewer 2 Report
The paper “Domain switching-based nonlinear coupling response for giant magnetostrictive materialsauthors” by Yunshuai Chen, Pengyang Li, Jian Sun and Guoqing Chen contributes to the development of magnetostrictive materials and devices based on them. The article is of high quality and contains interesting scientific results. I cannot fully appreciate the quality of some of the calculations provided, but the overall impression of the work is positive. Before the publication the text should be carefully reviewed for English and printing errors, because it contains some. In addition, there are some uncritical remarks:
1) Lines 69-70: “From a microscopic point of view, the nature of magnetostrictive materials is polycrystalline.”
Not necessary, it is possible to use magnetostrictive material in single crystal state.
2) Lines 118-119: “Magnetostrictive materials generate demagnetizing fields due to shape anisotropy and internal defects when subjected to applied magnetic fields.”
This is a very general statement and applies to all ferromagnetic materials, not just magnetostrictive ones.
3) Line 261. “…depends on the saturation magnetostriction λ100 and λ111.” Regarding to λ100 and λ111 it is better to use term “magnetostriction constants”.
4) Lines 261-262: “We assume both λ100, λ111 are equal to and saturation magnetostrictive λ”
Unclear sentence.
5) Caption of Fig. 3, Lines 457-459.
There are no units of compressive stress values.
6) Fig. 4.
It is necessary to make the same scale along the y-axis of both figures (a and b) for a better comparison of the results.
7) Lines 498-500. “Similarly, the rate of change of the magnetostriction coefficient λ for the magnetic field H, i.e., the permeability µm, becomes smaller with increasing applied mechanical compressive load in the lower and moderate axial magnetic field range.”
It is not quite correct to use the term permeability in this case. Moreover, the designation with the µm is commonly used for maximum permeability in magnetic materials. If dλ/dH is meant here, then it is better to use just such a designation, which will be much more understandable to the reader.
8) Lines 530-531. “The Young's modulus can approach the intrinsic Young's modulus Es…”
Authors should describe what they mean under intrinsic Young’s modulus
9) Lines 550-551. “Finally, for the practical use of Terfenol-D in rotating giant magnetostrictive ultra-550 sonic transducers where slicing is often used to reduce eddy current and hysteresis losses…”
Slicing reduces eddy current losses, but not hysteresis.
10) Lines 590-594. The last sentence before conclusions.
This one sentence occupy five lines and is very difficult to understand. Please rework it.

Before the publication the text should be carefully reviewed for English and printing errors, because it contains some (not so many).
Author Response

(The authors gave the same response as above.)

Reviewer 3 Report
To provide quantitative modeling of the induced magnetostrictive effects in giant magnetostrictive materials (Terfenol-D ), the authors propose a multilevel three-dimensional constitutive model based on a microscopically phenomenological approach taking into account the domain rotation mechanism, considering the interactions between elastic-inelastic strain and hysteresis effects at the grain, polycrystalline complex and macroscopic scales. The physical fields at larger spatial scales are obtained by amplifying the information of the corresponding fields at more minor spatial scales.
The authors use the experimental results reported by several authors to verify the magnetostrictive hysteresis results of the theoretical model for the same external environment (temperature) and different prestress excitation conditions. The results associated with the experiments show that the established model can reasonably predict the magnetostrictive response under different external mixed stimuli. It can provide theoretical guidance for the precise control of nonlinear vibrations and the optimal design of the rotating giant magnetostrictive transducers at both microscopic and macroscopic multiple scales.
The subject studied in this work is of great importance for applications and the obtained results are interesting for the readers of MDPI Materials journal.
The manuscript is well written, the original material is clearly presented, and conclusions are supported by the obtained results. However, I have some comments:
1. In the Introduction, the authors mostly cite references concerning the used models for evaluation of magnetostrictive materials properties. (please give full description of models, not abbreviations: SS, HT, ZL, JK, etc.). However, the cited literature is quite old, please give more recent citations (for example, Zhan, Y.S.; Lin, C.H. A constitutive model of coupled magneto-thermo-mechanical hysteresis behavior for Giant Magnetostrictive Materials. Mech. Mater. 2020, 148, 103477, https://doi.org/10.1016/j.mechmat.2020.103477; Daniel, L.; Domenjoud, M. Anhysteretic Magneto-Elastic Behaviour of Terfenol-D: Experiments, Multiscale Modelling and Analytical Formulas. Materials 2021, 14, 5165. https://doi.org/10.3390/ma14185165, etc.)
2. It would be useful to describe the main properties and application advantages of Terfenol-D material and compare with other magnetostrictive materials explaining why Terfenol-D was chosen for this study. How is critical chemical composition of Terfenol-D and what is optimal composition for applications? Table 1 lists different Ms from different literature sources, what is the reason (microstructure, chemical composition etc.)?
3. The authors state (line 55) that “…it has been noticed that these models, showing good nonlinear coupling agreement at low and medium applied magnetic fields, also show significant discrepancies between the predicted and experimental results at high magnetic fields and high prestress.“ It would be usefull to explain, why high magnetic fields are important (and how high?). What properties of material could change the saturation field and is it important for applications?
4. Line 454: “As shown in Fig. 3, the present constitutive model's predictions generally agree with the ZL model results at varying stresses at lower and higher magnetic fields.“ It is difficult to understand this statement from Fig.3. Please explain in more details. (For example, Fig.4 is clear and comparison of results is well demonstrated.)
5. Line 480: “Similarly, for comparison, we extract and optimize the experimental material parameters of Gao et al. for the new constitutive model prediction. Fig. 5 shows the magnetization and magnetostrictive hysteresis responses of Terfenol-D at varying ambient temperatures and stress states...“ It means that Fig. 5 presents modelling results. However, what information could be found from experiment and compared with theory and comparison remains unclear (only material parameters were taken from Gao et al.?). What was the purpose to extract only parameters without comparison?
6. Line 585: „Fig. 8 gives...“ Is it Fig 8 or 9?
Minor editing of English language is required. The authors have carefully to read the text.
Author Response

(The authors gave the same response as above.)

Reviewer 4 Report
I have reviewed the manuscript focused on modeling the non-linear response of a giant magnetostrictive actuator using continuum based formulation. I find the manuscript difficult to read as well as difficult to follow the model development logic of what the authors are doing as well as what they are trying to achieve. The authors use mixed variables and in many cases do not define variables being used. The article jumps around in the modeling space quite significantly and as such I find it challenging to formulate a positive opinion on this article.
English quality is OK
Author Response
Thank you very much for your kind criticism.

Round 2
Reviewer 1 Report
After revision the paper look more good and I recomend it to publication
Author Response
We greatly appreciate your guidance and help. We are sincerely thankful for all your comments and suggestions, which have significantly helped us improve the manuscript. This manuscript has been carefully revised based on these valuable suggestions.
Reviewer 2 Report
The authors revised the article in accordance with the comments of the reviewers. However, the paper still contains some flaws. In addition, some new ones appeared in added text. Thus, authors should carefully read and double-check the entire text again in order to eliminate all flaws and problems. Here are some examples:
1) Lines 74-75: “From a microscopic point of view, the nature of magnetostrictive materials can be used in poly-crystalline state”
2) Lines 269-270. “We assume that both λ100, λ111 are both equal to the saturation magnetostrictive…”
3) Lines 60-64. Added text. The first sentence of the two added is knocked out of the logical chain of text. It is not clear the purpose of mentioning this information. Further development of this idea and information about how the reversal phenomenon is related to the current work is necessary.
4) Equations (9) and (30). There is no decoding what is v.
5) Lines 558-562. What amorphous slices are?
7) The text still contains many large sentences (4-5 lines each), which are very difficult to read. Please change this to improve the readability of the text.

In general, the English language of the article is of a sufficient level. But there are some typos and bad wording which should be fixed. In addition, there are very long sentences (4-5 lines each) that are difficult to understand.
Author Response
The authors revised the article in accordance with the comments of the reviewers. However, the paper still contains some flaws. In addition, some new ones appeared in added text. Thus, authors should carefully read and double-check the entire text again in order to eliminate all flaws and problems. Here are some examples:
We greatly appreciate your guidance and help. We are sincerely thankful for all your comments and suggestions, which have significantly helped us improve the manuscript. This manuscript has been carefully revised based on these valuable suggestions.
Point 1: Lines 74-75: “From a microscopic point of view, the nature of magnetostrictive materials can be used in poly-crystalline state”. Lines 269-270. “We assume that both λ100, λ111 are both equal to the saturation magnetostrictive…”
Response 1: Thank you very much for your support and guidance. We have thoroughly followed your suggestion to improve the manuscripts fully.
Point 2: Lines 60-64. Added text. The first sentence of the two added is knocked out of the logical chain of text. It is not clear the purpose of mentioning this information. Further development of this idea and information about how the reversal phenomenon is related to the current work is necessary.
Response 2: We are grateful for the suggestion. We have revised it.
Point3: Equations (9) and (30). There is no decoding what is v.
Response 3: We are grateful for the suggestion. We have revised it
Point 4: Lines 558-562. What amorphous slices are?
Response 4: Thank you for the suggestion. We have revised it.
Point 5: The text still contains many large sentences (4-5 lines each), which are very difficult to read. Please change this to improve the readability of the text.
Response 5:Thank you for this valuable suggestion. Therefore, the manuscript's language has been thoroughly revised and edited according to the reviewer's comments in this version, so we hope it can meet the journal's standards. Thank you very much for your helpful comment.

Reviewer 3 Report
The authors corrected the manuscript according to reviewer comments, so it could be accepted in present form.
English quality is sufficient.
Author Response

(The authors gave the same response as above.)
